# Continual BERT: Continual Learning for Adaptive Extractive Summarization of COVID-19 Literature

**Jong Won Park**
Deerfield Academy
jpark21@deerfield.edu

## Abstract

The scientific community continues to publish an overwhelming amount of new research related to COVID-19 on a daily basis, leading to many literature without little to no attention. To aid the community in understanding the rapidly flowing array of COVID-19 literature, we propose a novel BERT architecture that provides a brief yet original summarization of lengthy papers. The model continually learns on new data in online fashion while minimizing catastrophic forgetting, thus fitting to the need of the community. Benchmark and manual examination of its performance shows that the model provide a sound summary on new scientific literature[1].

## 1 Introduction

The rapid emergence of the novel coronavirus without much known history has engrossed the international scientific community, resulting in an overwhelming amount of publications and data released on a daily basis. The rate of publications has far exceeded the time-consuming peer-review process, leaving many important information with little to no attention. In an attempt to absorb and utilize the unprecedented amount of COVID-19 scientific literature, prominent journals have opened publications to the public and several platforms have prompted the data science community to aid in the process. One of the notable platform has been COVID-19 Open Research Dataset (CORD-19)[2] containing thousands of papers published on PubMed and multiple tasks to understand the papers.

Recent progress on language processing has made possible the exploration of massive text corpus otherwise infeasible by manual work.

Attention-based mechanisms (Vaswani et al., 2017) and pre-trianed language representations such as BERT (Devlin et al., 2018), Open GPT-2 (Radford et al., 2019), XLNet (Yang et al., 2019), and ELMo (Peters et al., 2018) have achieved a great success in many language fields, including sentence prediction and text summarization. Many language models are adopting a common practice of pre-training on a huge corpus mined from the web, followed by a fine-tuning process targeted for specific tasks. Following this trend, we focus on utilizing the popular BERT architecture for text summarization task, more specifically extractive summarization, where important sentences are picked from the text verbatim. This task fits the need of the scientific community to rapidly process and extract important information from the inundating number of COVID-19 publications while adhering to their original text.

However, as COVID-19 papers are published on a daily basis, many of them with time-sensitive or unseen content, the model also needs to train in an online fashion without experiencing catastrophic forgetting. To this end, we propose *Continual BERT*, a novel BERT architecture built on existing techniques to learn and extract summaries from a continual stream of new tasks while retaining previously learned information. Heavily inspired by (Schwarz et al., 2018), our architecture utilizes two separate BERT models with layer-wise connections and deploys an alternating training process to minimize catastrophic forgetting. It also stacks a small Transformer encoder on top for extracting summary sentences from text.

## 2 Related Work

**Continual Learning** Recent efforts to train models online, where new data (tasks) flow in a time-sensitive, sequential manner, have increased to fit

---

[1]Our code is available at https://git.io/JJJf0
[2]semanticscholar.org/cord19

to the real-world training scenarios. Progressive Neural Networks (Rusu et al., 2016) instantiates new neural networks with layer-wise connections for new tasks, which mitigates catastrophic forgetting but renders inscalable. Progress & Compress (Schwarz et al., 2018) addresses the scalability by using two separate neural networks with layer-wise connections and online Elastic Weight Consolidation (Kirkpatrick et al., 2016). Dynamically Expandable Networks (Lee et al., 2017) takes a different approach by adaptively sizing the nuerons in each layer of a network, regularized with group sparse regularization (Scardapane et al., 2016).

Albeit the rapid development on continual learning, few works has focused on incorporating the process onto BERT for language processing. ERNIE 2.0 (Sun et al., 2019) modifies the pre-training aspect of BERT with a continual learning framework that learns on a broad spectrum of tasks. In contrast, our model modifies the fine-tuning process of BERT for continual learning, which enables leveraging any pre-trained models and focus more on the task-specific fine-tuning process.

**Extractive Summarization** Summarization of text has two categories: extractive and abstractive. The former extracts sentences deemed as a summary, while the latter constructs a unique summary that assimilates the extracted sentences. In the field of extractive summarization, *SummaRuNNer* (Nallapati et al., 2016) uses a sequential model based on Recurrent Neural Network, *LATENT* (Zhang et al., 2018) distingushes latent and activated variable sentences and extracts "gold" summaries the latter sentences to improve training, *NeuSum* (Zhou et al., 2018) jointly learns and scores extracted sentences, and *BertSum* (Liu, 2019) stacks Transformer layers on top of BERT for sentence extraction. Although our architecture assimilates BertSum, it differs in that it implements an additional component of continual learning for online tasks.

## 3 Continual BERT

### 3.1 Architecture

The base of *Continual BERT* is two identical BERT-base models - 12-layer, 768-hidden, and 12-heads - initialized with the pre-trained *bert-base-uncased* weights. Following Progress & Compress, the first BERT model is referred to as Active Column (AC) for its ability to train actively without restrictions, while the second BERT model is named Knowl-

| Dataset | Type | Annotated Articles |
|---|---|---|
| NCBI Diesase | Disease | 793 |
| CTD-Pfizer dataset | Drug | 18,410 |
| ScisummNet | General | 1,009 |
| CORD-19 | COVID-19 | 57,037 |

Table 1: Scientific literature dataset with either annotated abstracts or original summaries, including CORD-19 used for our model (as of June 27)

edge Base (KB) for its preservation of previously learned information.

The model trains using an alternative training scheme, which is described on 3.2. During the latter stage of the training, the model uses online Elastic Weight Consolidation (EWC) (Kirkpatrick et al., 2016) in conjunction with knowledge distillation. EWC calculates the Bayesian posterior distribution of parameters using Laplace approximation to calibrate gradient descent towards the overlapping learning region of both the previous and new tasks. The modified online version of EWC addresses the quadratic cost of the original EWC by using a running sum of the diagonal Fisher information matrix and the mean of online Gaussian approximation.

The model also establishes layer-wise connections from KB to AC using custom adaptors to enable positive forward transfer of previous tasks. Inputs are calculated in parallel through KB and AC, where KB captures the hidden states at each Transformer layer and wires it to the one layer up in AC, with calculations as follows:

$$h_i = \sigma(W_i h_{i-1} + \alpha_i \odot U_i \sigma(V_i h_{i-1}^{KB} + c_i) + b_i)$$

where $h_{i-1}$ and $h_{i-1}^{KB}$ are the hidden states from layer *i-1* of AC and KB, respectively, $W_i, U_i, V_i$ learnable weights, $b_i, c_i$ biases, $\sigma$ non-linear operation, $\odot$ element-wise multiplication, and $\alpha$ learnable vector. $\alpha$ is initialized from a uniform distribution on the interval $[0, 1)$.

For extracting summary sentences from literature, we stack a two-layered Transformer encoder on the top. More formally, a Transformer encoder (2-layer, 768-hidden, and 12-heads) computes the importance of each sentences in text with hidden states from the two BERT system.

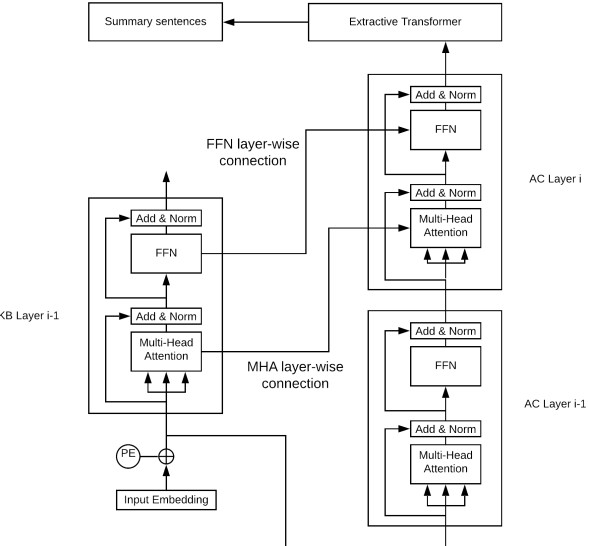

Figure 1: Figure 1. Architecture of *Continual BERT*. Two identical BERT models are initiated, and training on new tasks involves layer-wise connection from Knowledge Base to Active Column. The final output from Active Column is fed into a small, two-layered Transformer stack that outputs the extracted summary sentences. The newly learned parameters are consolidated into the previously learned parameters.

## 3.2 Training

Training is divided into two stages, compress and progress, which are executed in the order listed.

**Compress**   During compress, the first stage, AC trains normally on a new task with an exception of incorporating layer-wise connections from KB on each layer. This provides positive forward transfer from previously learned tasks to benefit training on the new task. No other restrictions are imposed other than the loss function, hence "unrestricted". The model aims to minimize the loss:

$$H_p(q) = -\frac{1}{N}\Sigma_{i=1}^{N}y_i * \log(p(y_i))$$
$$+ (1 - y_i) * log(1 - p(y_i)$$

where $y$ is the abstract sentence (label).

**Progress**   After compress, progress begins, where knowledge from AC is distilled into KB in a teacher-student scheme with online EWC to preserve previous information and thus minimize catastrophic forgetting. The model aims to minimize the EWC loss:

$$L_{EWC} = \frac{1}{2}\|\theta - \theta_{k-1}^*\|_{\gamma F_{k-1}^*}^2$$

where $\theta$ is the parameters learned on a new task, $F_{k-1}^*$ and $\theta_{k-1}^*$ the diagonal Fisher and the mean

(of all previous learned parameters) of the online EWC Gaussian approximation, and $\gamma$ hyperparameter to dictate the degradation of previously learned parameters. In addition, the model aims to reduce the knowledge distillation loss:

$$L_{KD} = \mathcal{E}[KL(\pi_k(\dot{|}x)||\pi^{KB}(\dot{|}x))]$$

where, $x$ is the input, $\pi_k(\dot{|}x)$ and $\pi^{KB}(\dot{|}x)$ prediction of AC (after learning) and KB, $\mathcal{E}$ expectation over a task dataset. In conclusion, during the progress tep, the model minimizes the loss:

$$L_{progress} = L_{EWC} + L_{KD} \qquad (1)$$

The alternating training procedure allows the model be online - continuously learning on new tasks. Online training is essential to actively updating COVID-19 literature, where new information not only relies on previous data but also are time-critical to the understanding of the status quo of the coronavirus.

Since our tasks contain text that are similar in topic, i.e. COVID-19, we preserve the weights of AC throughout new tasks to benefit from previous unrestricted learning. If the tasks differ in nature, we recommend re-initializing AC with pre-trained BERT weights to maximize the learning of the new task.

| Model | ROGUE-1 F | ROGUE-2 F | ROGUE-L |
|-------|-----------|-----------|---------|
| Continual BERT | 31.6 | 12.7 | 30.5 |
| BertSum | 33.0 | 13.4 | 31.6 |
| SummaRuNNer | 24.2 | 8.9 | 10.1 |

Table 2: Evaluation of *Continual BERT* and other extractive models on ScisummNet dataset (Yasunaga et al., 2019) using ROUGE metric.

## 3.3 Dataset

We use PubMed articles from CORD-19 as our dataset **??**. We modify the dataset to include only articles with abstracts, resulting in total of 42,000 out of 57,037 articles. We use abstracts as the gold summary and train the model to extract up to 15 sentences that are most similar to the abstract. To simulate an online training environment, we divide the articles into tasks of 5,000 articles, ordered by ascending publication time. The model initially learns on tasks with older articles and gradually transits through to the newest articles in an online fashion.

We prepare the dataset as a classification task for BERT, with each classification being a sentence, as proposed by (Liu, 2019). Each sentence is padded with *[CLS]* at the front to tag it as a classifiable entity and alternating word embedding scheme is used to distinguish adjacent sentences as different sentences. The model learns to classify *[CLS]*, which outputs indice for summary sentences.

## 3.4 Experiment

The models weights are initialized with Xavier initialization (Glorot and Bengio, 2010). For optimization, we use AdamW (Loshchilov and Hutter, 2019) with weight decay of $0.01$, $\beta_1 0.9$, $\beta_2 0.999$, and $\epsilon 1e - 6$. For learning scheduler, we use linear learning schedule with learning rate of $5e - 4$ and $5\%$ of each task (approximately 300 articles) for warm-up; for knowledge distillation, we use $\tau$ of $2.0$ and $\alpha_{ce}$ $0.5$; for EWC, we use $\lambda$ of 15 and $\gamma$ $0.99$. Other settings include weight decay of $0.01$, batch size of $64$, and 3 epoch for training AC and KB on each task.

## 4 Results

After learning on nine tasks ordered in ascending publication time, *Continual BERT* recorded a loss of $0.21$ for compress (consolidation) stage and $2.15$ for progress stage, indicating a difficulty in calibrating to both the previous and new parameters. ROUGE evaluation on the ScisummNet is presented on 2 and manual evaluation on recent COVID-19 literature is presented on **??**. The manual summary evaluation, which is a more realistic and sound technique compared to ROGUE, shows that the model can produce a sound summary of extracted sentences spread throughout the literature. This summary assimilate many sentences provided by the authors, which further supports the model's capability to learn well on new tasks in an online manner.

## 5 Discussion

The online training ability of *Continual BERT* enables adaptive learning on new data flowing in a time-sequential manner, especially fitting to the overwhelming amount of COVID-19 literature published on a daily basis. In contrast to the provided abstracts, extractive summarization of those literature can provide not only a sound, original summary of the article but also indications of where the interesting sentences and ideas lies within the text. This feature can be handy with longer papers, much of COVID-19 literature, as the readers can save significant amount of time while understanding the broad idea of the papers. The scalable architecture of *Continual BERT* also enables continually learning over longer time and in more frequency to digest new research data and information faster.

The difficulty that *Continual BERT* experienced during the progress step can be justified with the fact that CORD-19 contains publications dating back to the 20th century, which present radically different information to the more modern publications. This information disparity can be fixed by penalizing more for older publications through time threshold, such as before the coronavirus pandemic.

We hope that the model provides a ground for other researchers to explore into the area of summarization for COVID-19 and many other literature. For future explorations, we propose constructing a dynamic version of the model, such as dynamically increasing/decreasing network neurons.

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

# A  Manual evaluation

Manual evaluation on a recent COVID-19 publication published on July 1, 2020. Extracted summary is in lower-case since the pre-trained model is uncased (*bert-base-uncase*).

### Title

Structure of the full SARS-CoV-2 RNA genome in infected cells (Lan et al., 2020) (28 pages)

### Abstract

SARS-CoV-2 is a betacoronavirus with a single-stranded, positive-sense, 30-kilobase RNA genome responsible for the ongoing COVID-19 pandemic. Currently, there are no antiviral drugs or vaccines with proven efficacy, and development of these treatments are hampered by our limited understanding of the molecular and structural biology of the virus. Like many other RNA viruses, RNA structures in coronaviruses regulate gene expression and are crucial for viral replication. Although genome and transcriptome data were recently reported, there is to date little experimental data on predicted RNA structures in SARS-CoV-2 and most putative regulatory sequences are uncharacterized. Here we report the secondary structure of the entire SARS-CoV-2 genome in infected cells at single nucleotide resolution using dimethyl sulfate mutational profiling with sequencing (DMS-MaPseq). Our results reveal previously undescribed structures within critical regulatory elements such as the genomic transcription-regulating sequences (TRSs). Contrary to previous studies, our in-cell data show that the structure of the frameshift element, which is a major drug target, is drastically different from prevailing in vitro models. The genomic structure detailed here lays the groundwork for coronavirus RNA biology and will guide the design of SARS-CoV-2 RNA-based therapeutics.

### Extracted Summary

sars-cov-2 is an enveloped virus belonging to the genus beta coronavirus, which also includes sarscov, the virus responsible forthe 2003 sars outbreak, and middle east respiratory syndrome coronavirus (merscov), the virus responsible for the 2012 mers outbreak. despite the devastating effects these viruses have had on public health and the economy, currently no effective antivirals treatment or vaccines exist. there is therefore an urgent need to understand their uniquerna biology and develop new therapeutics against this class of viruses. coronaviruses (covs) have single - stranded and positive - sense genomes that are the largest of all known rna viruses (27 − 32 kb) (masters , 2006). previous studies oncoronavirus structures have focused on several conserved regions that are important forviral replication. for several of these regions, such as the 5' utr, the 3' utr , and the frameshift element (fse), structures have been predicted computationally with supportive experimental data from rnase probing and nuclear magnetic resonance (nmr) spectroscopy (plant et al. , 2005; yang and leibowitz , 2015).