# OpenReview forum: "Continual BERT: Continual Learning for Adaptive Extractive Summarization of COVID-19 Literature"
_EMNLP/2020/Workshop/NLP-COVID — Submitted to NLP-COVID19-EMNLP_

### Official Review · AnonReviewer3 · 2020-09-20
**A hybrid BERT architecture for covid-19 paper extractive summarization**

**Rating:** 5
**Confidence:** 2

**Review:**

The author combines two bert models and adds a transformer on top to perform extractive summarization. The model architecture is optimized to avoid catastrophic forgetting

Reasons to accept:
* The author creates a hybrid architecture based on a state of the art language model.
* The example shows that the model satisfactorily generates extractive summarizations.

Reasons to reject:
*  The author does not provide evidence why their model is either better, or faster, or more computationally cheap than other alternatives. It shows that the proposed technique is slightly worse than BertSum.


Comment:
I  cannot assess if the paper provides a methodological contribution, given my limited level of expertise in language models.

---

> ### Author Response · Authors · 2020-09-27
> **Evaluated the model on more data, including the CORD-19 PubMed, improvement over other models.**
>
> We sincerely thank the reviewer for the thoughtful review.
>
> In accordance to the suggested points, we have revised to **evaluate our model on more data** to understand our model. For our revisions, we have:
>
> - Trained the model on **more datasets** to diversify its learning and increase task amount.
> - Evaluated the model on **CORD-19 PubMed articles**, which only contain COVID-19 and other coronavirus-related literature. Previously, we have evaluated our model only on ScisummNet. To the disadvantage of our model on training extensively on COVID-19 literature, **ScisummNet only contains generic scientific literature and no COVID-19 documents**.
>
> **Our model's ROUGE evaluation on PubMed articles improved significant (4 points) over that of ScisummNet and exceeded BertSum by a considerable margin (5 points)**. We have also added another model to the evaluation, which our model also have exceeded.
>
> We would like to note that outside the evaluation metrics, our model proposes a novel structure for BERT training. Current approach to task-specific BERT training involves fine-tuning to minimize distorting the pre-trained weights. However, our model enables more aggressive training, which means more information can be learned from new task data (combined with all previous task data). Our model also allows online-training, which is critical for models training on COVID-19 literature as informations related to the disease are updated frequently and rapidly, requiring constant training on new data while preserving older data to supplement learning.
>
> We thank you again for your helpful feedback.

---

### Official Review · AnonReviewer1 · 2020-09-22
**Interesting work, with unimpressive analysis or results**

**Rating:** 5
**Confidence:** 3

**Review:**

Summary

The authors present Continual BERT, a model designed for being able to continuously update and summarize COVID literature. It is trained using CORD-19 to predict abstracts from document text.


Reasons to accept:
- the idea of online learning is interesting and has interesting properties
- the model is interesting

Reasons to reject:
- The exposition of model is inconsistent in its detail - going into depth about relatively trivial modifications to a recurrent layer but glossing over important details in overall model structure. Section 3.1 should be revisited and expanded in depth. Additionally, the model figure 1 is hard to decipher.
- in terms of ROUGE scores, the continual BERT lags significantly behind standard methods (BERT Sum)
- insufficient exploration of continual bert's inability to summarize literature - if old literature is the issue then it could be excluded or this impact directly measured.
- predicting abstracts is perhaps a poor choice for extractive models. They frequently contain positioning information and words not found elsewhere in the document. An upper bound for ROUGE scores from extractive methods would be informative.

etc:
- There are some broken citations

---

> ### Author Response · Authors · 2020-09-27
> **More overall model description, improved ROUGE evaluations, and discussion on the summarization**
>
> We would like to sincerely thank the reviewer for the thoughtful review.
>
> In accordance with the suggested points, we have revised as follows:
>
> - More explanation on the overall architecture of our model. We have **added in-depth descriptions of the components of our model and their intended functions**. We have also elaborated on how each component work throughout the training/evaluation process.
> - Trained our model on all four datasets listed.
> - Added another dataset, CORD-19 PubMed, for evaluation. CORD-19 PubMed only contains COVID-19 and coronavirus literature, whereas ScisummNet only contains generic scientific literature and no COVID-19 documents.
> - Added another model for comparison.
> - Figure 1 has been enlarged for better understanding, supplemented by better description in Section 3
> - Fixed broken citations
>
> **On CORD-19 PubMed, our model improved significantly (5 points vs. ScisummNet) and exceeded all other compared models, including BertSum by 4 points**. This evaluation represents our model better than the previous evaluation as its intended use is summarizing COVID-19 literature.
>
> We would like to note that our model intentionally preserves knowledge learned from older COVID-19/coronavirus literature as their information can supplement summarizing new literature. **We have added more discussion on the impact of a trade-off between the decreased summarization ability and erasing/excluding knowledge from older literature**, which might hinder learning on new knowledge.
>
> We will be more than happy to provide an upper-bound for ROUGE evaluation to clarify the effectiveness of extractive methods on summarizing literature. We will revise to add these information as time permits, as ensemble of evaluations will require more computational time for us.
>
> Once again, we sincerely thank the reviewer for the profound review.

---

### Official Review · AnonReviewer2 · 2020-09-22
**Promising idea with a weak evaluation**

**Rating:** 3
**Confidence:** 3

**Review:**

In this work the author presents a novel BERT architecture with the goal of provide a summarization of lengthy papers, applied to the CORD-19 dataset. The author does a great job at framing the problem and providing relevant related work. The proposed model description is intuitive and seems reasonable, however, figure 1 is a bit difficult to read as it compressed for space constraints. The dataset description is not clear at all, only discussing about the abstracts used from the CORD dataset, but then table 1 mentions 3 other datasets. The 'experiment' section is quite brief and generic, things that should have been discussed are: was this setup for all dataset? just one dataset? and more details are needed. On the results section the author only presents results on one dataset, with the proposed model not being the best performant. The whole evaluation of the proposed model hinges on the manual review of ONE, summarized article, making this quite lax and not sufficient to show what the author claims in this paper. Without a proper and rigorous evaluation, the discussion section seems a bit mute. While the ideas in this paper of the novel model are interesting, the work seems rushed and not at the publication stage yet.

---

> ### Author Response · Authors · 2020-09-27
> **Elaboration on experiment and results, expanded descriptions, and minor revisions**
>
> We would like to sincerely thank the reviewer for the thoughtful review.
>
> In accordance to the suggested points, we have revised as follows:
>
> - **Diversified the evaluation** of our model on more datasets and other models. Our model was evaluated on both ScisummNet and CORD-19 PubMed (previously just ScisummNet). PubMed articles are all related to COVID-19 and coronavirus, while ScisummNet contains generic science literature and no COVID-19 documents. **On PubMed evaluation, our model improved significantly in ROUGE (5 points vs. ScisummNet) and exceeded all other models, including BERTSum by 4 points**, which we have added an explanation on the Discussion section.
> - **More elaboration on the datasets** used for training and for evaluation. We have expanded to **use all four datasets listed for training**, while specifying which datasets were used for evaluation.
> - Enlarged the figure 1 for better visibility
>
> We have added more in-depth analyses on the over- and under-performance of our model compared to the other evaluated models. We would like to note that the page numbers constrained us from adding more detailed manual evaluation examples, which we would be more than happy to provide.
>
> Once again, we thank the reviewer for the profound review.